# Green Mining Takes Place at the Power Plant

**Zhiyi Zhang** [1,2,3,*], **Hao Liu** [2,*], **Hui Su** [2] **and Qiang Zeng** [1]

1   School of Ecology and Environment, Xinjiang University, Urumchi 830046, China; zengqiang@xju.edu.cn
2   School of Geology and Mining Engineering, Xinjiang University, Urumchi 830046, China;
    suhui054710@163.com
3   Collaborative Innovation Center of Green Mining and Ecological Restoration for Xinjiang Mineral Resources,
    Urumchi 830046, China
*   Correspondence: xjuzhyiyi@163.com (Z.Z.); liu_xj2017@163.com (H.L.); Tel.: +86-0991-211-1408 (H.L.)

**Abstract:** The number of large coal power plants, characterized by pithead plants, is increasing rapidly in major coal mining countries around the world. Overburden movement caused by coal mining and greenhouse gas emissions caused by coal thermal power generation are intertwined, and have become important challenges for mine ecological environment protection at present and in the future. In order to provide more options for green mining in large coal power plants, a large coal power base in northwest China was taken as the researching background in this paper, and a green mining model considering the above two aspects of ecological environment damages was proposed; that is, the carbon dioxide greenhouse gas produced by coal-fired power plants can be geologically trapped in goaf, whose overburden stability is controlled by backfill strips made of solid mine waste. In order to explore the feasibility of this model, the bearing strength of the filled gray brick consisting mainly of aeolian sand and fly ash under different curing methods was firstly studied, and it was discovered that the strength of the gray brick significantly improved after carbonization curing. After that, X-ray diffraction (XRD) and scanning electron microscopy (SEM) were employed to compare the mineral composition and its spatial morphology in gray brick before and after carbonization, and it is believed that the formation of dense acicular calcium carbonate after carbonization curing was the fundamental reason for the improvement of its bearing strength. Finally, a series of stope numerical models were established with UDEC software to analyze the surface settlement, crack propagation height and air tightness of the overlying strata, respectively, when goaf was supported by the backfilling strips with carbonized gray brick. The research results of this paper showed that the stability of overlying strata in goaf can be effectively controlled by adjusting the curing methods, width and spacing of the filled gray brick, so as to facilitate the following geological sequestration of carbon dioxide greenhouse gas in goaf. Consequently, the ecological environment damages caused by coal mining and utilization in a large coal power base can be resolved as a whole, and the purpose of green mining can be achieved as desired.

**Keywords:** green mining; low-carbon utilization; backfill mining; carbon dioxide storage; gray brick

## 1. Introduction

For several decades, coal has significantly contributed to global energy needs, accounting for 25% of global energy production in 2000, 30% in 2010 and 27% in 2020 [1]. However, in the process of coal mining, it often brings about serious overburden movement, which causes the loss of groundwater resources and surface collapse, changes the soil structure and seriously damages the ecological environment [2,3]. For example, in China, the area of land destroyed by mining has reached 2 million hm$^2$ [4], in which the area of settlement land has reached 1 million hm$^2$ [5]. These mining impacts pose significant environmental, socio-economic and mining layout challenges. For this reason, these adverse impacts and mitigation measures have been extensively studied in several countries, including Russia [6], Australia [7,8], the United Kingdom [9], South Africa [10,11], India [12,13] and

Germany [14]. Meanwhile, the number of large coal power plants, characterized by pithead plants, is increasing rapidly in major coal mining countries around the world. Overburden movement caused by coal mining and greenhouse gas emission caused by coal thermal power generation are intertwined, and have become an important challenges for mine ecological environment protection at present and in the future, as shown in Figure 1.

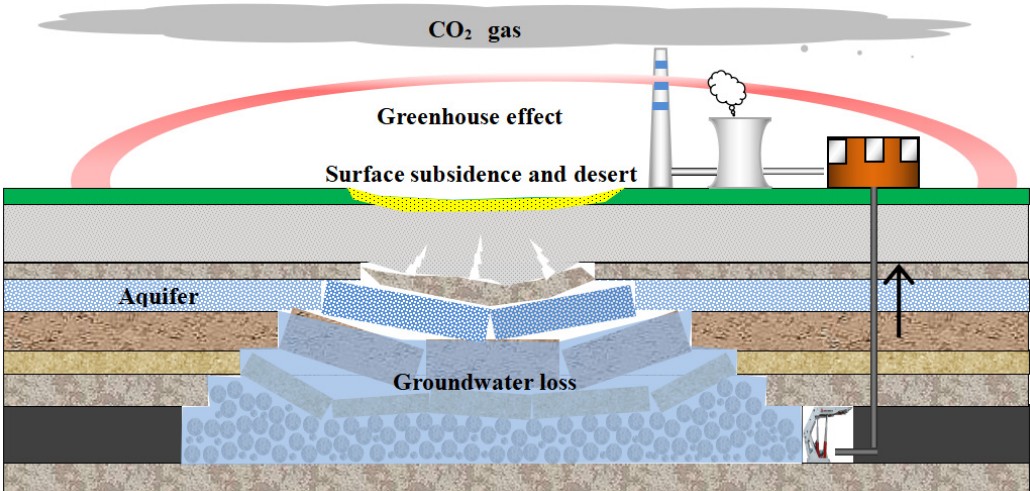

**Figure 1.** Ecological and environmental effects of the traditional coal exploitation and utilization mode in large coal power bases.

To resolve the problems of the environmental damage caused by the mining and utilization of coal as a fossil fuel in large coal power bases, an innovative mode of green mining and low-carbon utilization of the coal resources is proposed, as shown in Figure 2. Firstly, the aeolian sand abundant on the surface of mining area and fly ash produced by thermal power plant are used as the main raw materials to make the embryo body of filling gray brick. Then, the gray brick is curved with carbon dioxide gas from the thermal power plant and backfilled in the underground gob to support overlying strata. Finally, the carbon dioxide gas can be injected and stored in the gob.

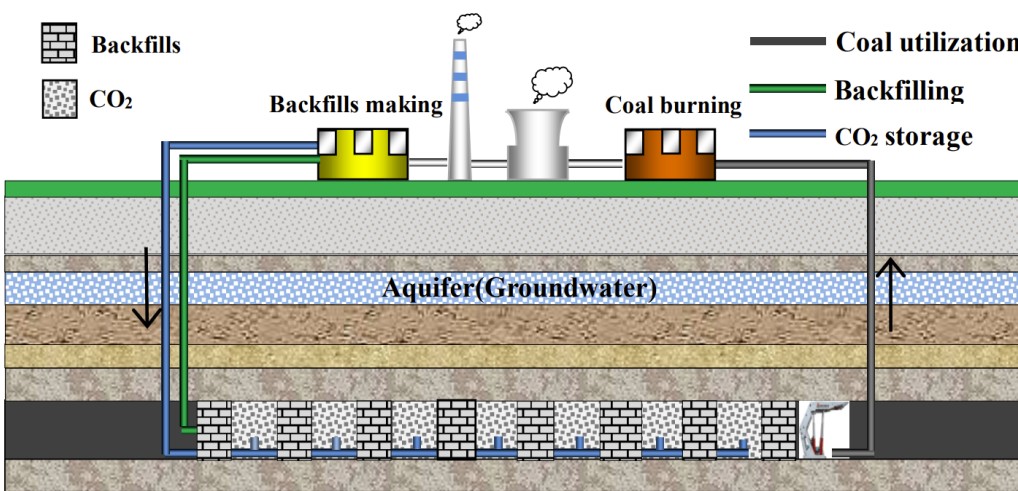

**Figure 2.** Innovative mode of green mining and low-carbon utilization for coal resources in large coal power bases.

In the past few decades, backfilling has been extensively applied in underground coal mining to resolve safety and environmental problems. The backfill bodies were employed to reduce ground subsidence [15,16], underground water-loss [17,18] and so forth. Meanwhile, backfill mining employs coal gangue (CG), fly ash (FA) and tailings as primary materials

to provides a feasible approach for treatment and utilization of solid wastes from mining industries [19,20]. On this basis, many valuable studies have been carried out recently. For the first time, a mixture of classified tailings with cement began to be tested in 1969 at the Mount Isa mine in Australia. This technology has been successfully applied in many mines in Canada and Sweden. In Russia (USSR), tailings were first used to prepare a monolithic backfill in 1969 at the Riddersky mine. In 1970, one chamber was laid at the Gaisky mine (USSR). Currently, man-made wastes are widely used in the production of backfill composite around the word. Technogenic waste replaces not only the inert filler, but also the binder. Ercikdi et al. used waste glass and silica fumes as artificial pozzolanas to prepare tailings cementing filling materials and investigated the effect of the type of waste materials on the early strength and later strength of the filled object [21,22]. Feng et al. studied the mechanical properties of cemented filling materials in which gangue was partially replaced by waste concrete [23,24]. Peyronnard and Benzaazoua studied the effect of CAlSiFrit and deinking sludge fly ash as partial binder replacement in cemented paste backfill (CPB) [25]. Cihangir et al. used alkali-activated neutral and acidic blast furnace slags (AASs) with aqueous sodium silicate (LSS), and sodium hydroxide (SH) were tested as alternative binders to OPC for CPB of high-sulfide mill tailings [26]. Deng et al. developed a new type of cemented filling material, using waste rock as a coarse aggregate, FA as a fine powder, slag as an activator and ordinary Portland cement as a binder [24,27]. Xu et al. studied the strength development and microstructure evolution of cemented tailings backfill containing different binder types and contents [28]. Zhou et al. explored the feasibility of replacing cementitious filling materials with air-accumulated sand as aggregate, and investigated the effects of fly ash (FA) content, cement content, lime slag (LS) content and concentration on the mechanical properties of air-accumulated sand-based cementitious filling materials [24]. Wang et al. used hydrogen peroxide ($H_2O_2$) as a chemical blowing agent to improve the foaming performance of the cemented foam and the reinforcing effect of the foam based on the cemented foam with gangue and fly ash as the main raw materials [29]. Ermolovich et al. studied the possibility of creating and using nanomodified backfill material based on the waste from enrichment of water-soluble ores [30].

Analyzing the above, it can be noted that reducing or ablating the damage to the ecological environment caused by mining activities green mining is a very topical issue. With the centralized mining and utilization of coal resources, related environmental problems will appear at the same time, which provides a possibility for the comprehensive solution of various problems. Therefore, the purpose of this study is exploring a green mining model that simultaneously controls the strata stability over goaf and geologic sequestration of carbon dioxide greenhouse gas in goaf. To achieve this goal, the following tasks need to be addressed: (1) how to prepare the backfilling material in goaf with certain mine solid waste, and provide it with the bearing strength to meet the requirements through appropriate curing method; (2) how to evaluate the stability of the overlying strata after goaf is backfilled by the above-designed materials.

## 2. Materials and Methods

### 2.1. Experimental Materials

The cementing agents used in the experiments were quicklime and gypsum. The quicklime was produced by the Tianshan Cement Factory in Xinjiang, China. The main composition of quicklime is CaO, but it also contains a small amount of $SiO_2$ and MgO; its CaO content reaches up to 89%, as shown in Figure 3a,d. The gypsum was purchased from the factory as β-type hemihydrate gypsum ($2CaSO_4 \cdot H_2O$). The main chemical composition of gypsum includes CaO and $SO_3$, and it also contains a small amount of $SiO_2$, $Al_2O_3$ and $Fe_2O_3$, as shown in Figure 3b,e. The fly ash used in the experiments was class II fly ash produced by the Hongyanchi Power Plant in Urumqi, Xinjiang, China. The average particle size of the fly ash is 0.035 mm. The fly ash is primarily composed of mullite and quartz; the $SiO_2$ content is 55% and $Al_2O_3$ content is 25%, as shown in Figure 3c,f. The aggregate

used in the experiments was aeolian sand from the Shanshan Desert, Xinjiang, China, with a particle size primarily between 0.1–0.25 mm and a non-uniformity coefficient of 1.82.

**Figure 3.** Experimental materials. (**a**) XRD pattern of quicklime; (**b**) XRD pattern of gypsum; (**c**) XRD pattern of fly ash; (**d**) chemical element content of quicklime; (**e**) chemical element content of gypsum; (**f**) chemical element content of fly ash. Here, XRD indicates X-ray diffraction analysis.

## 2.2. Specimen Preparation and the Experimental Study

In this study, the ratio of water: cement was 2:3, and the ratio of gypsum: fly ash: quicklime: aeolian sand was 1:2:3:12. The dry ingredients were mixed and thoroughly homogenized using a blender (HJW-60 series) for 3 min before water was added into the mixture, and then stirred for another 2 min. After mixing, the mixed slurry was cast into an iron mold with dimensions of 5 cm × 5 cm × 5 cm and three specimens were made for each curing age using different curing methods to reduce the test error. The specimens were then subjected to natural, autoclave, carbon dioxide and autoclave–carbon dioxide curing for 4 days, 8 days, 12 days and 16 days [31], as shown in Table 1. Finally, uniaxial compressive strength (UCS) tests, carbonization tests, X-ray diffraction analysis (XRD) tests and scanning electron microscope (SEM) tests were carried out on the specimens, as shown in Figure 4.

The UCS tests on gray brick were carried out using the SANS brand hydraulic single shaft compressor in the Mechanics Laboratory, School of Mechanical Engineering, Xinjiang University, China, as per the Chinese standard (JGJ/T 70-2009). The maximum test force of a uniaxial compressor is 300KN, and the accuracy of the force value is below ±0.3%. According to the standard, a displacement loading model was used to avoid specimens rapidly breaking [32]. In this way, the whole stress–strain curve was obtained. In this study, the pre-peak loading speed was 0.1 mm/s and the loading speed after the peak was 0.2 mm/s. The XRD tests were carried out using a D8 Advance X-ray powder diffractometer (Bruker AXS GmbH), and the radiation source of this instrument is Cu target. The scanning range 2θ is 5°~80° and the scanning speed is 10°/min. The SEM tests were carried out using a LEO-1430VP scanning electron microscope (Zeiss, Oberkochen, Germany) with the magnification of 50~20,000 times. Carbonation depth of gray brick under different curing methods was detected using the phenolphthalein alcohol method (phenolphthalein alcohol

solution with mass fraction of 1% as chromogenic agent) [33], and carbonation depth was measured by digital carbonation depth scale (China Zhuolin Science and Technology, Beijing, China), with measuring accuracy of 0.01 mm and measuring range of 0–25 mm.

**Table 1.** Specimens' curing method.

| Specimen Number | Curing Age | Curing Methods |
|---|---|---|
| I-A-1, I-A-2, I-A-3<br>I-B-1, I-B-2, I-B-3<br>I-C-1, I-C-2, I-C-3<br>I-D-1, I-D-2, I-D-3 | 4 days<br>8 days<br>12 days<br>16 days | After 12 h resting, the specimens were demolded and cured in a humid and ventilated natural environment. |
| II-A-1, II-A-2, II-A-3<br>II-B-1, II-B-2, II-B-3<br>II-C-1, II-C-2, II-C-3<br>II-D-1, II-D-2, II-D-3 | 4 days<br>8 days<br>12 days<br>16 days | After 12 h resting, the specimens were placed directly on the inner surface of the autoclave equipment without demolding. After 5 h of autoclave curing in an environment of 0.165 Mpa and 130 °C, the specimens were taken out and then demolded in a humid and ventilated natural environment for curing. |
| III-A-1, III-A-2, III-A-3<br>III-B-1, III-B-2, III-B-3<br>III-C-1, III-C-2, III-C-3<br>III-D-1, III-D-2, III-D-3 | 4 days<br>8 days<br>12 days<br>16 days | After 12 h resting, the specimens were demolded and cured in drying dishes with $CO_2$ concentration of 0.25 mol/L and pressure of 718.98 Pa. |
| IV-A-1, IV-A-2, IV-A-3<br>IV-B-1, IV-B-2, IV-B-3<br>IV-C-1, IV-C-2, IV-C-3<br>IV-D-1, IV-D-2, IV-D-3 | 4 days<br>8 days<br>12 days<br>16 days | After resting for 12 h, the specimens were placed cured in the autoclave equipment for 5 h, and then the specimens were taken out and demolded in drying dishes with $CO_2$ concentration of 0.25 mol/L and pressure of 718.98 Pa for curing. |

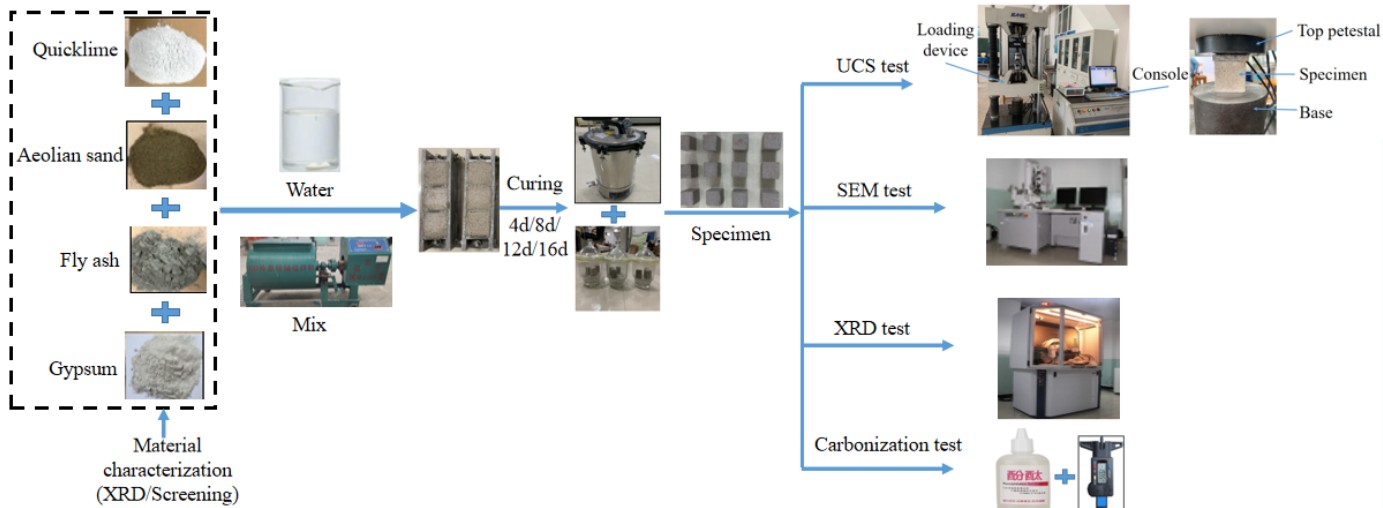

**Figure 4.** Experimental process. Here, SEM indicates scanning electron microscope.

*2.3. Study on the Overburden Control Effect of Gray Brick in Backfill Mining*

2.3.1. Establishing a Numerical Model

UDEC is a two-dimensional numerical calculation program based on a continuum simulation discrete element, which mainly simulates the mechanical behavior of discontinuous medium (such as joint block) under static or dynamic load conditions. UDEC discontinuous media is reflected by the combination of separated blocks, and the joints are treated as boundary conditions between blocks, allowing blocks to move and turn along the joint surface [34]. UDEC can clearly simulate the development of cracks in overlying strata during gray brick backfill mining, so as to achieve the desired simulation effect. Because large amounts of coal resources are distributed in western China with shallow buried depths, large thicknesses and thin overlying rock strata, the stratum conditions in the mining area in western China were simplified into three components: a bedrock layer, a weak cementation layer and loose overburden (Figure 5a) [35,36]. Based on this, a typical

numerical model was established. The block division of the different coal strata in the model is shown in Figure 5b. The established model was 500 m in length and 340 m in height, and the heights of the floor, coal seam, roof, bedrock layer, weak cemented layer and loose overburden were 22 m, 8 m, 6 m, 94 m, 100 m and 100 m, respectively. The bottom boundary of the model was fixed, the surrounding boundary was displacement constrained and the upper boundary was free. The Mohr–Coulomb yield criterion was used to calculate the constitutive relationship of the block. The strata joints were simplified into horizontal and vertical joints, and the surface contact Coulomb slip model was adopted. Boundary coal pillars of 100 m were reserved on the left and right sides of the model to eliminate the boundary effect, and the actual advance length was 300 m. The mechanical parameters of each rock layer are shown in Table 2.

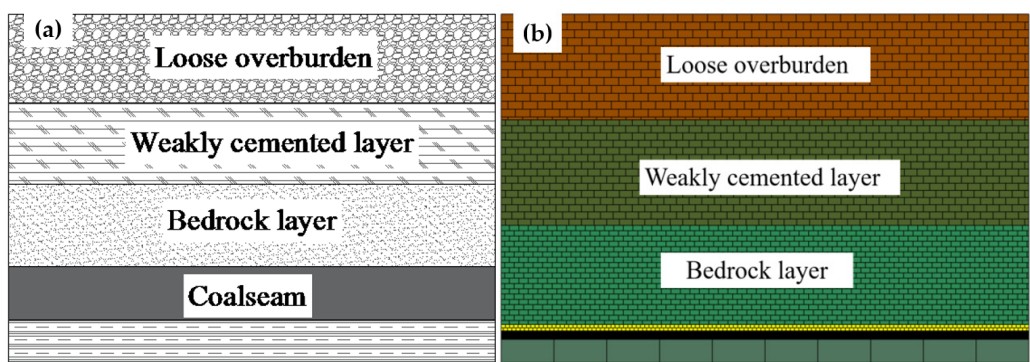

**Figure 5.** UDEC numerical model. (**a**) Typical stratum conditions in the Xinjiang mining area; (**b**) numerical model.

**Table 2.** Mechanics parameters of the rock strata used in simulation.

| Stratum | Density Kg/m$^3$ | Bulk Modulus /GPa | Shear Modulus /GPa | Friction Angle /Degree | Cohesion /MPa | Tensile Strength /MPa |
| --- | --- | --- | --- | --- | --- | --- |
| Loose overburden | 2200 | 0.05 | 0.03 | 25 | 0.7 | 0 |
| Weakly cemented layer | 2580 | 1.03 | 7.5 | 44 | 3.5 | 2 |
| Bedrock layer | 2700 | 2.52 | 1.6 | 48 | 6.97 | 5.4 |
| Roof | 2700 | 2.52 | 1.6 | 48 | 6.97 | 5.4 |
| Coal seam | 1470 | 7.9 | 5.5 | 37 | 3.02 | 3 |
| Floor | 1700 | 5.15 | 4 | 36 | 2.21 | 2.26 |

### 2.3.2. Simulation Scheme

The full-height mining method was adopted, and the advance step of the working face was 10 m in this simulation. In addition, the mechanical parameters of the gray brick after 16 days of autoclave–carbon dioxide curing were used as the simulation parameters. The calculated time step for each step from the retrieval was determined to be 2000 steps based on the time effect of the site. Considering the filling equipment and the actual situation at mining sites, the top connection rate was set to 95% for the simulation. The specific simulation scheme is shown in Table 3.

**Table 3.** Simulation scheme.

| Simulation Variables | | Expected Simulation Results |
| --- | --- | --- |
| Different filling spacing | Different filling strip width | Surface subsidence and height of crack propagation |
| 10 m, 15 m, 20 m, 25 m, 30 m, 35 m | 11 m, 12 m, 13 m, 14 m, 15 m, 16 m | |

## 3. Results and Discussion

### 3.1. Analysis of the Bearing Strength of the Gray Brick

Typical uniaxial compression total stress–strain data of gray brick under different curing methods were selected for the analysis and are plotted in Figure 6a–d. It can be seen that, under different curing methods, the shapes of the total stress–strain curves of the gray brick are basically the same and the plastic characteristics of the gray brick do not change. The compressive deformation curves of the gray brick are similar to that of typical rock and experience four stages throughout the entire stress and strain process, namely the pore compaction stage (the concave curve), the elastic deformation stage (the oblique line), the plastic ring breaking stage (the concave curve) and the post-peak failure stage (the post-peak curve).

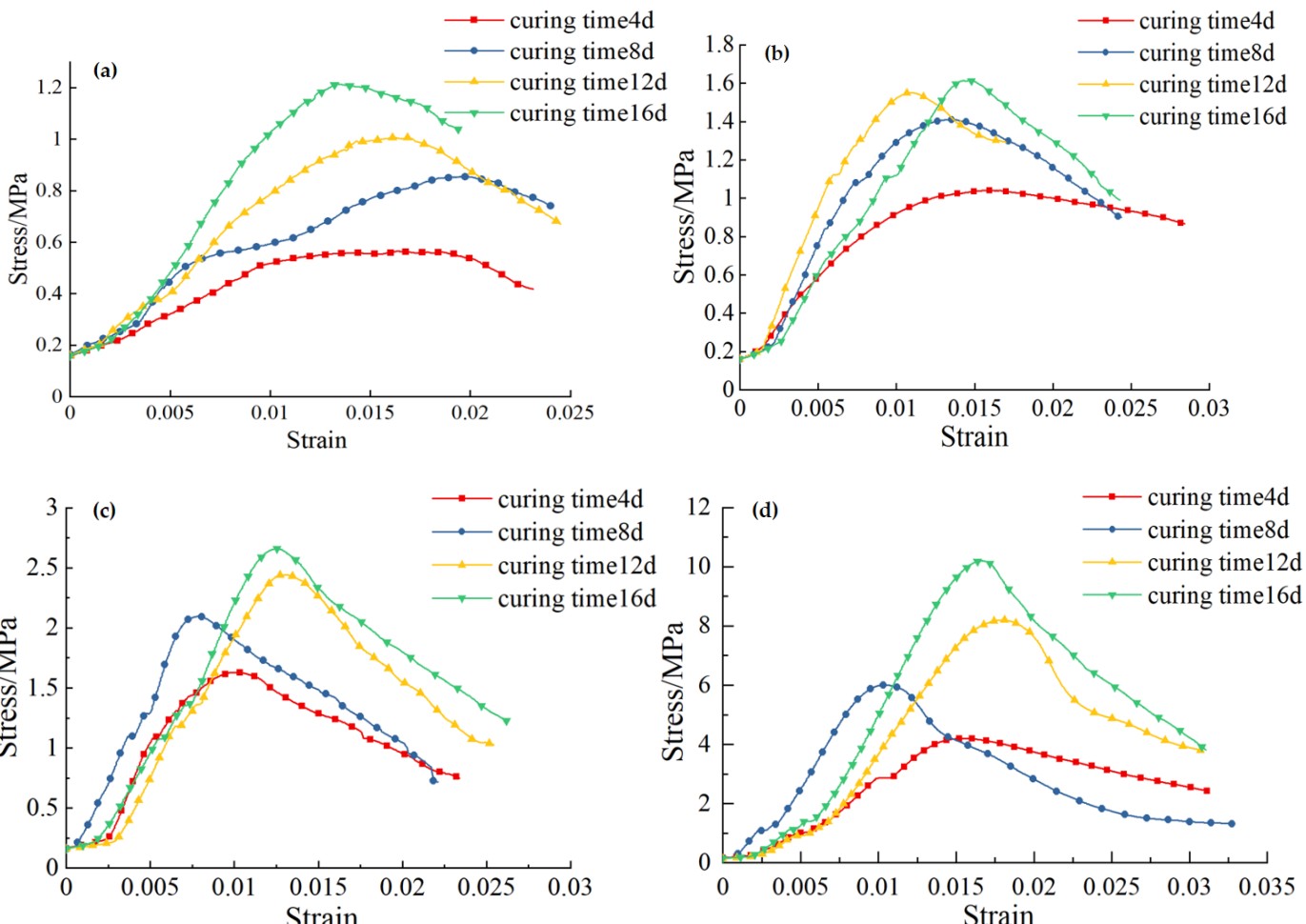

**Figure 6.** Complete stress–strain curves of gray brick under different curing methods. (**a**) Natural curing; (**b**) autoclave curing; (**c**) carbon dioxide curing; (**d**) autoclave–carbon dioxide curing.

The uniaxial compressive strengths of the gray brick under different curing methods and curing ages are shown in Figure 7. The results indicate that, under autoclave curing, carbon dioxide curing and autoclave–carbon dioxide curing, the bearing strength of the gray brick increased to different degrees compared with that of natural curing. The uniaxial compressive strength of the gray brick under 16 days of natural curing was 1.25 MPa, while the uniaxial compressive strengths of gray brick under 16 days of autoclave curing, carbon dioxide curing and autoclave–carbon dioxide curing were 1.58 MPa, 2.58 MPa and 9.65 MPa, respectively, increasing by 26%, 98% and 668%, respectively, compared with natural curing.

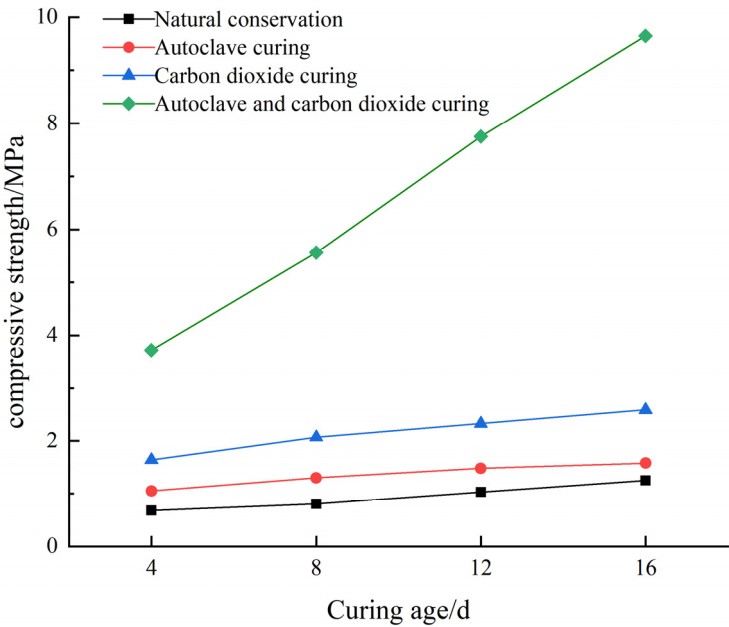

**Figure 7.** Bearing strengths of gray brick under different curing methods.

To study the failure modes of gray brick under different curing methods, typical failure surfaces under the different curing methods were selected to draw crack maps, as shown in Figure 8a–d. The results indicate that, under the different curing methods, there were two main cracks running through the samples, accompanied by several microcracks on the surfaces of the gray brick; these cracks had the same characteristics for all methods. Therefore, changing the curing method did not change the failure mode of the gray brick; all brick showed typical X-shaped conjugate shear failure.

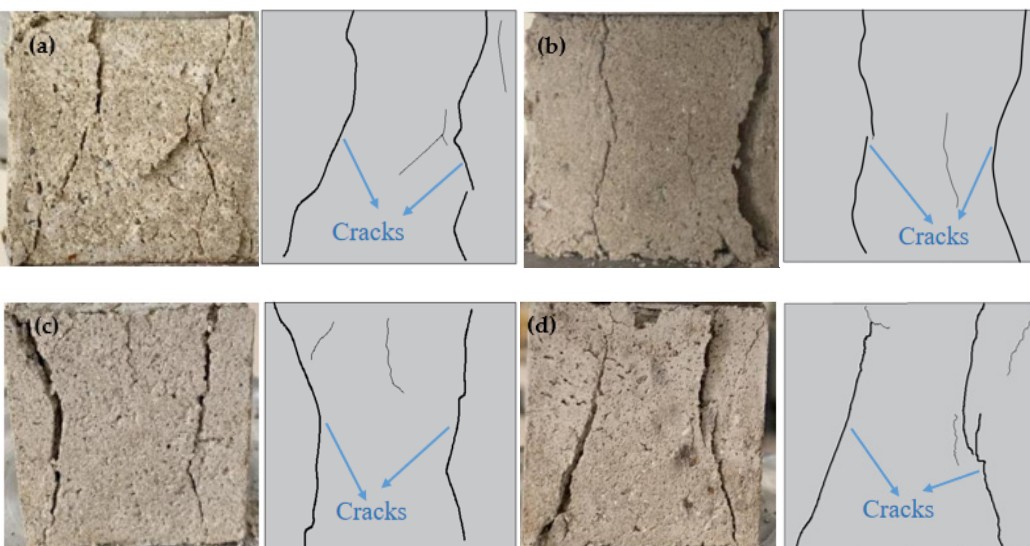

**Figure 8.** Failure modes of the gray brick under different curing methods. (**a**) Natural curing; (**b**) autoclave curing; (**c**) carbon dioxide curing; (**d**) autoclave–carbon dioxide curing.

### 3.2. Carbonation Curing Mechanism Analysis of the Gray Brick

The carbonization degrees of the gray brick under different curing methods were measured and the test results were analyzed, as shown in Figure 9a,b. It can be seen in Figure 9b that, under natural and autoclaved curing conditions, carbonization of the gray brick basically did not occur. The carbonization depths of the gray brick under 16 days of carbon dioxide curing and autoclave–carbon dioxide curing were 4.1 mm and

8.9 mm, respectively. Obviously, the carbonization degree of the gray brick was promoted by autoclave curing and, with increasing curing age, the carbonization rate of the gray brick slowed.

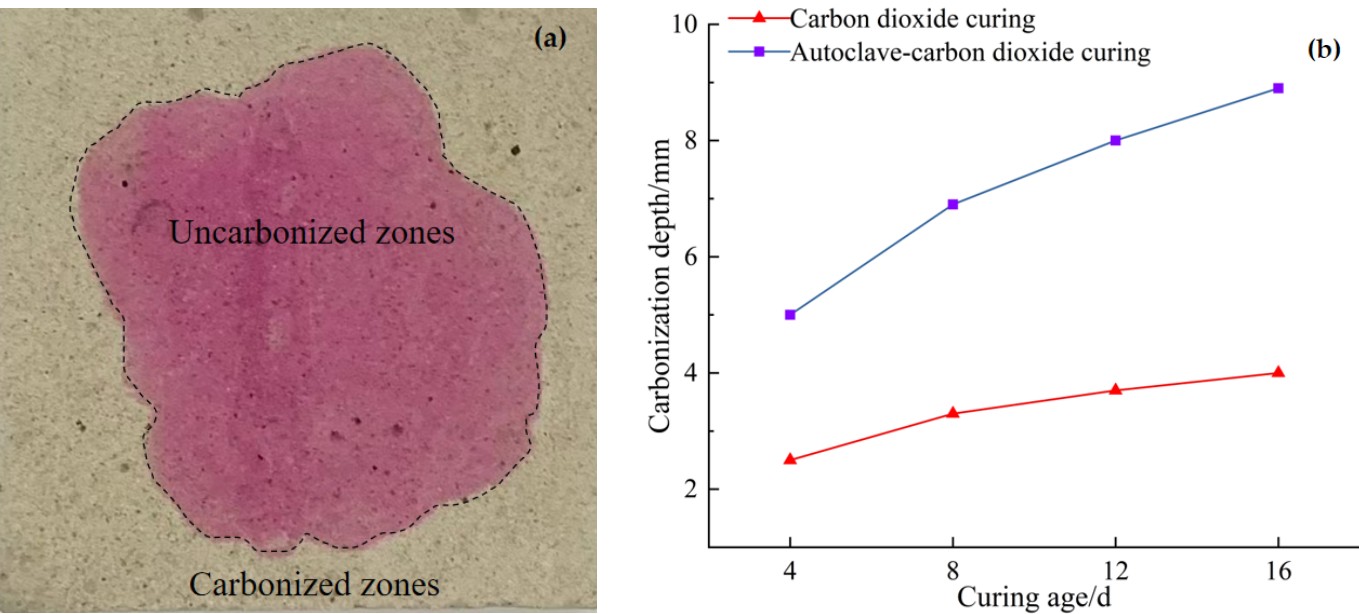

**Figure 9.** Carbonization depths of gray brick under different curing methods. (**a**) Carbonation diagram of specimen III-A-2; (**b**) carbonation depths of gray brick.

To study the mechanism of the increasing bearing strength of gray brick following carbonization, the internal hydration products and internal structure of the uncarbonized and carbonized parts of specimen III-A-2 were analyzed via XRD and SEM, as shown in Figure 10a–d. It can be seen that the internal compactness of the uncarbonized gray brick was small and that they contained a lot of gaps and pores; therefore, the bearing strength was primarily provided by the friction between the small internal particles. The main hydration products of the carbonized gray brick were $CaCO_3$ and $CaSO_4(H_2O)_2$, rather than $Ca(OH)_2$. In addition, the internal gaps and pores of the uncarbonized gray brick were filled by interlaced and needle shaped $CaCO_3$, resulting in higher compactness and integrity.

### 3.3. Stability Analysis of the Overlying Strata Filled with Gray Brick Strips

The simulation results showed that the maximum surface subsidence always occurred directly above the goaf, and this subsidence under mining with different filling spacings and filling strip widths is shown in Figure 11. It can be seen that the maximum surface subsidence reached a maximum of 0.675 m when the filling spacing was 35 m and the width of the filling strip was 11 m and that it reached a minimum of 0.403 m when the filling spacing was 10 m and the width of the filling strip was 16 m. The maximum surface subsidence could be reduced obviously when the filling spacing decreased from 35 m to 20 m and the filling strip width increased from 11 m to 13 m. In order to describe the quantitative relationship between the maximum surface subsidence (*S*) and the filling strip spacing (*x*) and the filling strip width (*y*) more accurately, the Gauss2D function of the Origin software was used to obtain the fitting relationship between above parameters, as shown in Equation (1). According to the results of fitting calculation, the fitting correlation coefficient ($R^2$) was 0.9926, the mean square error (MSE) was $3.86 \times 10^{-5}$ and the root mean square error (RMSE) was 0.0062, and the goodness of this fit was acceptable.

$$S = 59.3 - 58.9e^{-\frac{(x+4.7)^2}{421362} - \frac{(y-17.9)^2}{34322}} \ (x : 10\text{–}35 \text{ m}; \ y : 11\text{–}16 \text{ m}) \tag{1}$$

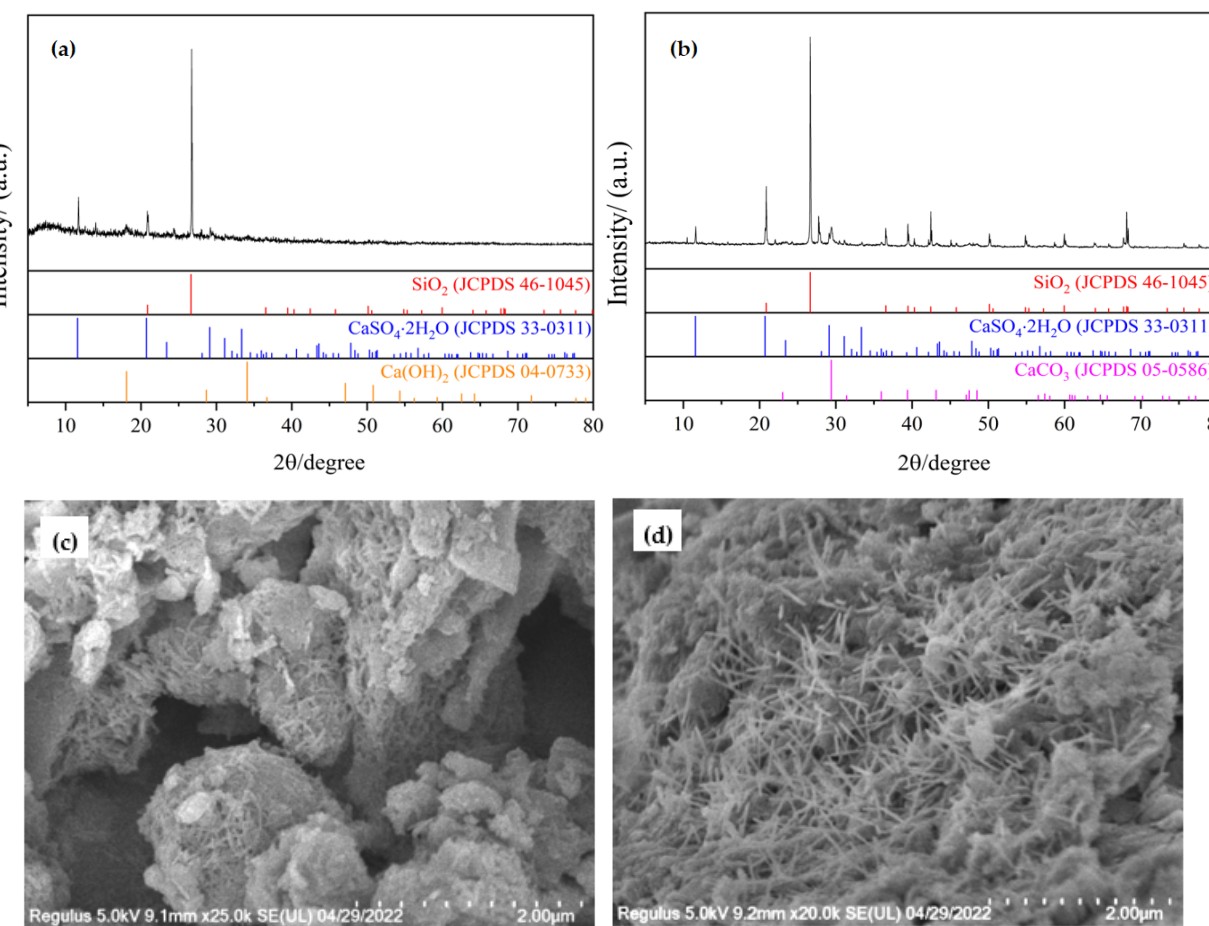

**Figure 10.** Comparison of the products and structures in the uncarbonized and carbonized parts of specimen III-A-2. (**a**) XRD of the uncarbonized part; (**b**) XRD of the carbonized part; (**c**) SEM of the uncarbonized part; (**d**) SEM of the carbonized part.

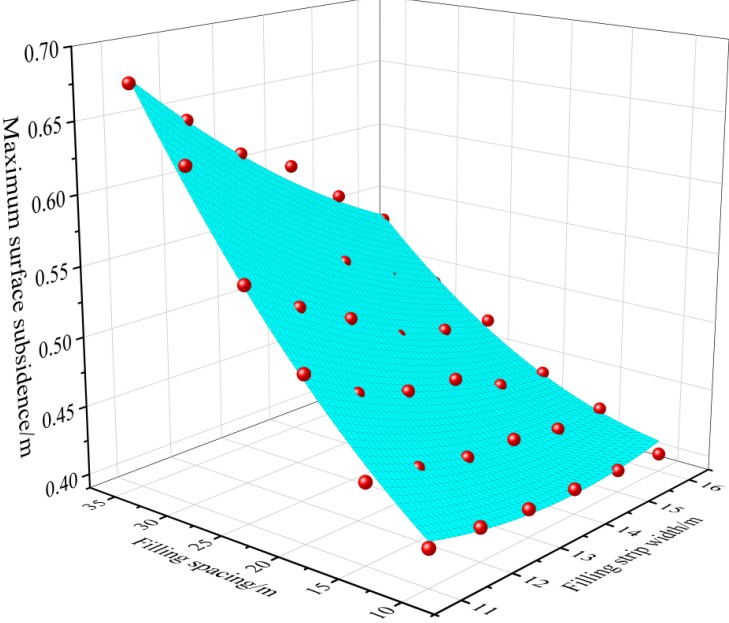

**Figure 11.** Fitting surface of the maximum surface subsidence with different filling spacings and filling strip widths.

According to Equation (1), the maximum surface subsidence (*S*) has an exponential relationship with filling strip spacing (*x*) and filling strip width (*y*). Further, the maximum surface subsidence (*S*) decreases continuously with decreasing filling spacing (*x*) and increasing filling strip width (*y*), which provides a valuable reference for controlling the maximum surface subsidence (*S*) through adjusting the backfill settings. The primary consideration for the filling strips spacing (*x*) is to avoid the caving of the immediate roof strata over the gob due to fact that the gob must be kept integrated for the purpose of following storage of carbon dioxide gas. The filling strip width (*y*) is commonly limited by the construction speed of filling strips because the backfilling speed and mining speed need to be consistent to ensure the continuity of backfill mining operations. In practice, the filling strips spacing (*x*) can be adjusted in a larger range compared with the filling strip width (*y*) because of the fact that the time for backfilling in a backfill mining cycle is always limited. Consequently, it is more feasible to control the maximum surface subsidence (*S*) by adjusting the filling strip spacing (*x*). It also should be noted that the parameters except for filling strip spacing (*x*) and filling strip width (*y*) in Equation (1) are only applicable to the engineering geological conditions in this study, and should be adjusted according to the mining depth, rock strata mechanics and backfilling materials' strength in different cases. It can be determined that the relationship between the maximum surface subsidence (*S*) and filling strip spacing (*x*) and width (*y*) described in Equation (1) could represent the general surface subsidence of underground mines in northwest China with the overburden composed of loose layer, weakly cemented layer and bedrock layer from top to bottom.

In the simulation process, if the joint contact appears to slip and have tensile failure, this indicates that the contact has fractured; that is, that cracks appear in the overburden [37]. The overburden fracture height under different filling spacings and filling strip widths is shown in Table 4. Based on the overburden fracture heights under different filling settings obtained from Table 4, the fracture propagation height (*H*) variation diagram of the overburden with different filling spacing (*x*) and filling strip widths (*y*) are drawn, as shown in Figure 12. In order to describe the quantitative relationship between the fracture propagation height (*H*) and the filling strip spacing (*x*) and the filling strip width (*y*) more accurately, the Gauss2D function of the Origin software was used to obtain the fitting relationship between above parameters, as shown in Equation (2). According to the results of fitting calculation, the fitting correlation coefficient ($R^2$) was 0.99938, the mean square error (MSE) was 15.2, and the root mean square error (RMSE) was 3.897 and the goodness of this fit was acceptable.

$$H = 30.9 + 719294\mathrm{e}^{-\frac{(x-109.1)^2}{1352} - \frac{(y+28.2)^2}{327.7}} \quad (x:\ 10\text{–}35\ \mathrm{m};\ y:\ 11\text{–}16\ \mathrm{m}) \tag{2}$$

According to Equation (2), the fracture propagation height (*H*) has an exponential relationship with filling strip spacing (*x*) and filling strip width (*y*). Further, the fracture propagation height (*H*) decreases continuously with decreasing filling spacing (*x*) and increasing filling strip width (*y*), which provides a valuable reference for controlling the fracture propagation height (*H*) through adjusting the backfill settings. It also should be noted that the parameters except for filling strip spacing (*x*) and filling strip width (*y*) in Equation (2) are only applicable to the engineering geological conditions in this study, and should be adjusted according to the mining depth, rock strata mechanics and backfilling materials' strengths in different cases. It can be determined that the relationship between the fracture propagation height (*H*) and filling strip spacing (*x*) and width (*y*) described in Equation (2) could represent the general fracture distribution of underground mines in northwest China with the overburden composed of loose layer, weakly cemented layer and bedrock layer from top to bottom.

**Table 4.** Fracture propagation height of different filling spacings and filling strip widths.

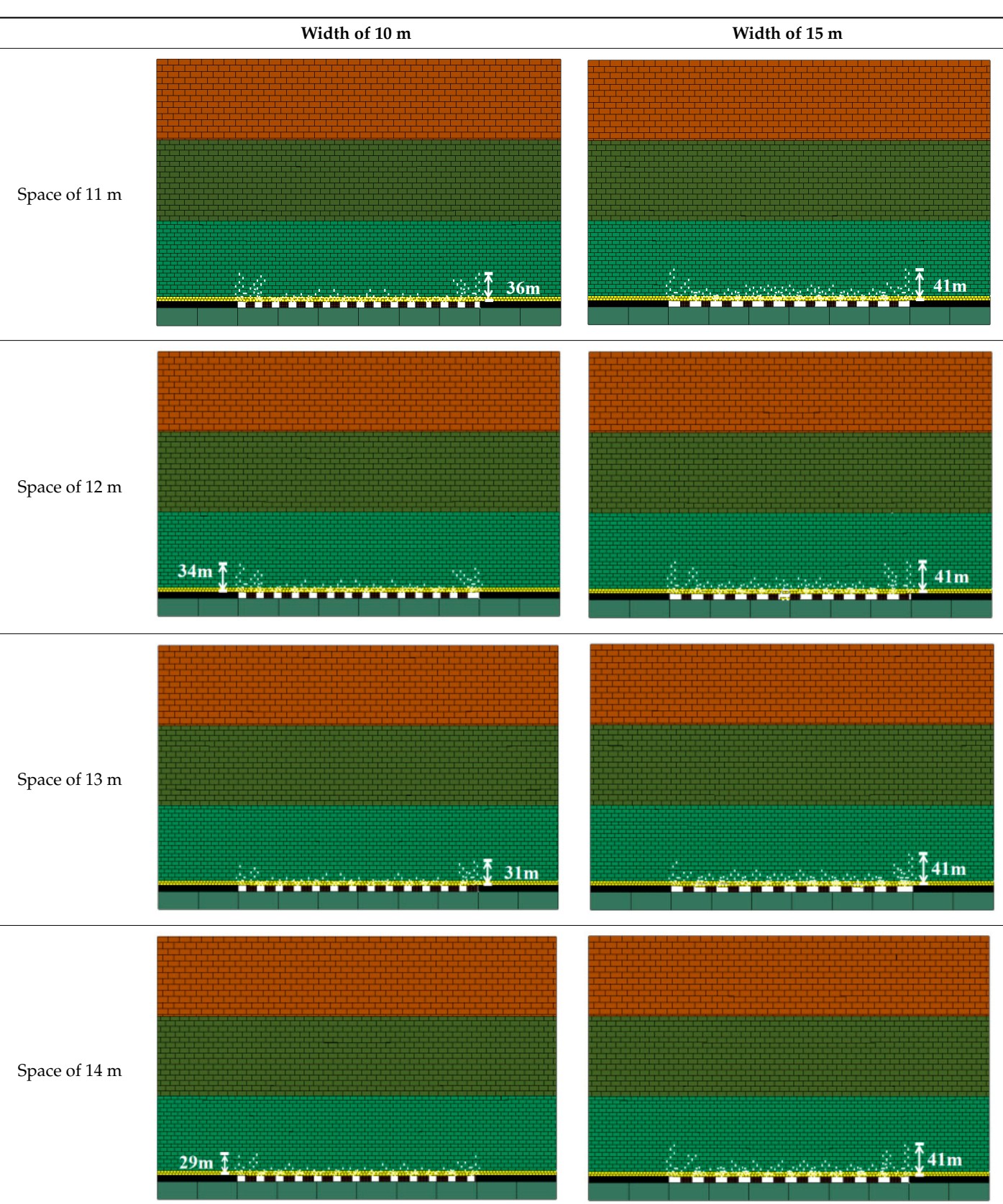

**Table 4.** *Cont.*

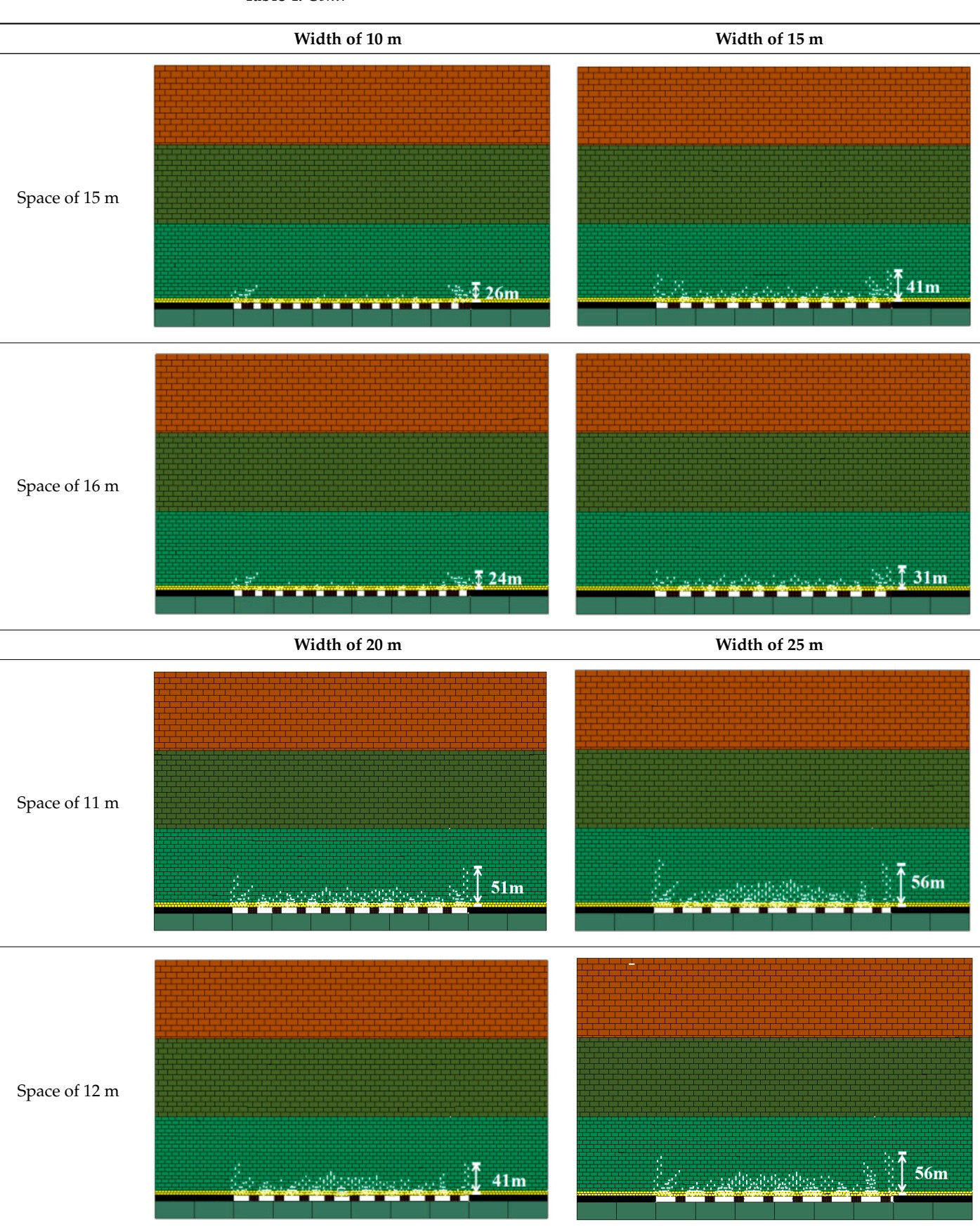

**Table 4.** *Cont.*

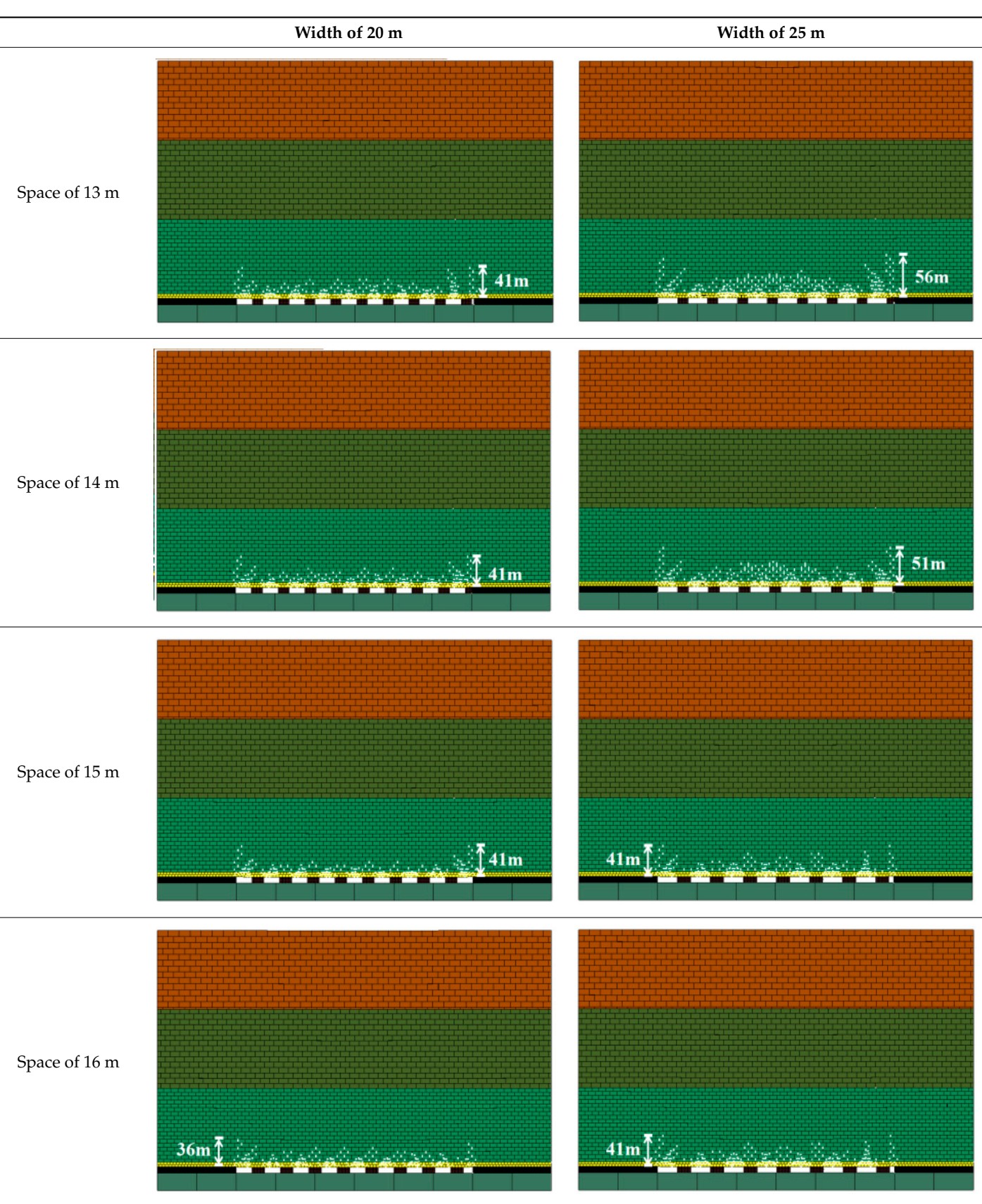

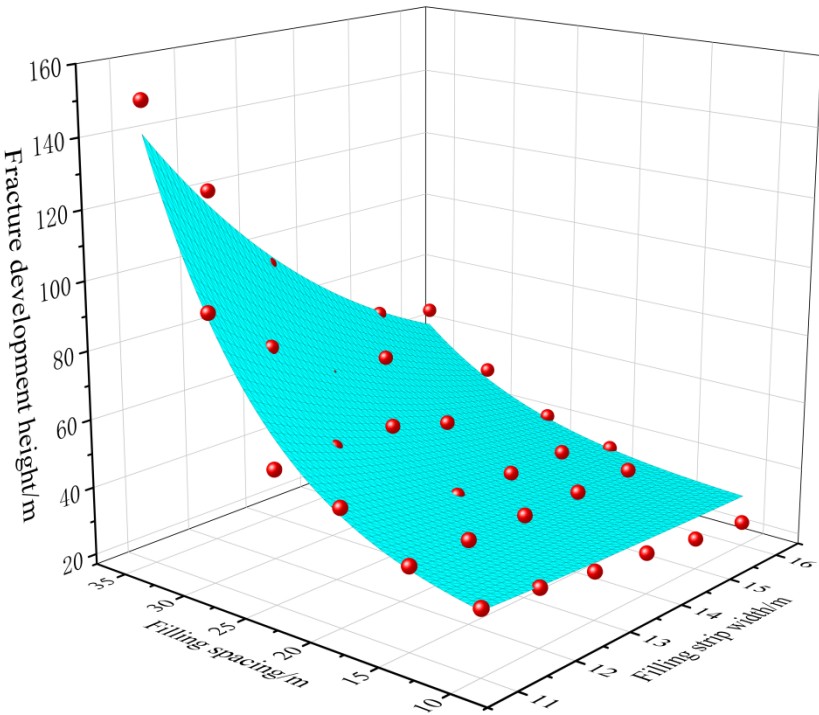

**Figure 12.** Fitting surface of different filling spacings, filling strip widths and overburden fracture propagation heights.

### 3.4. Evaluation of the Gas Tightness of Overlying Rock Filled with Gray Brick Strips

There are two main aspects that need to be considered with respect to the escape of $CO_2$ from goaf. The first aspect is that, when the capping pressure is less than the $CO_2$ sealing pressure, the formation stress is redistributed; this is primarily reflected in an increase in the capping cracks and permeability, resulting in the escape of the sequestered $CO_2$ [38,39]. Because of the depth of the coal buried in mining areas in western China, the $CO_2$ storage pressure in the goaf is small; therefore, it is difficult to reach the cap rock breakthrough pressure. Accordingly, this was not studied in this paper. The second aspect is that of the permeability of $CO_2$ through the micropores of the cap as a result of the difference in the pressure gradient, which is primarily related to the permeability coefficient of the cap. In general, the permeability of the bedrock layers is in the range of $10^{-1}$–$10^{-3}$ $\mu m^2$ because of the high internal microporosity; the permeability of the weakly cemented layer is generally relatively low at $10^{-11}$–$10^{-5}$ $\mu m^2$. Moreover, gas seepage is a slow and long-term process that can be calculated using Darcy's law:

$$Q = KA(P_1{}^2 - P_2{}^2)/2P_0\mu L \tag{3}$$

where $Q$ is the gas flow through rock, $cm^3/s$; $A$ is the cross-sectional area of gas passing through the rock, $cm^2$; $\mu$ is the Viscosity of gas, MPa·s; $P_1$ and $P_2$ are the rock inlet and outlet gas pressure, MPa; $L$ is the Length of the rock, cm; $K$ is the Permeability coefficient of gas to rock, $\mu m^2$; and $P_0$ is Atmospheric pressure, MPa.

Using Darcy's law, the $CO_2$ escape amount under different filling spacings and filling strip widths in this numerical model was calculated, where the strike length of the model working face was set to 200 m, the $CO_2$ sequestration pressure was set to 1 MPa, the permeability of the bedrock layer was set to $10^{-3}$ $\mu m^2$ and the permeability of the weakly cemented layer was set to $10^{-5}$ $\mu m^2$ combined with the actual situation of the goaf. The calculated results are shown in Figure 13. In order to describe the quantitative relationship between the gas escape quantity ($Q$) and the filling strip spacing ($x$) and the filling strip width ($y$) more accurately, the Gauss2D function of the Origin software was used to obtain the fitting relationship between above parameters, as shown in Equation (4). According

to the results of fitting calculation, the fitting correlation coefficient ($R^2$) was 0.99679, the mean square error (MSE) was 3.4, and the root mean square error (RMSE) was 1.84, and the goodness of this fit was acceptable.

$$Q = 6.58 + 786.5e^{-\frac{(x-109.1)^2}{703.9} - \frac{(y+21.9)^2}{282.3}} \quad (x: \ 10\text{–}35 \ \text{m}; \ y: \ 11\text{–}16 \ \text{m}) \tag{4}$$

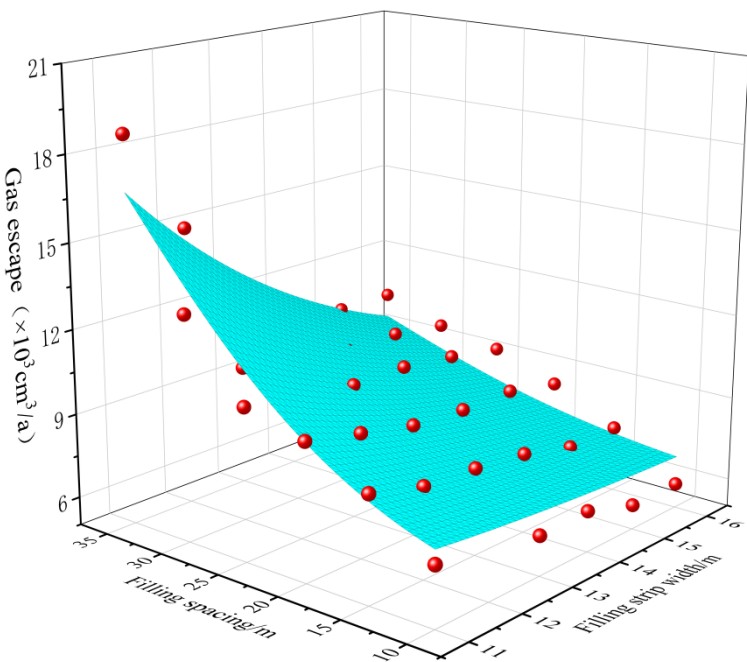

**Figure 13.** Fitting surfaces of different filling spacings and strip filling widths with the gas escape amount.

According to Equation (4), the gas escape quantity ($Q$) has an exponential relationship with filling strip spacing ($x$) and filling strip width ($y$). Further, the fracture gas escape quantity ($Q$) decreases continuously with decreasing filling spacing ($x$) and increasing filling strip width ($y$), which provides a valuable reference for controlling the gas escape quantity ($Q$) through adjusting the backfill settings. It also should be noted that the parameters except for filling strip spacing ($x$) and filling strip width ($y$) in Equation (4) are only applicable to the engineering geological conditions in this study, and should be adjusted according to the mining depth, rock strata mechanics and backfilling materials' strengths in different cases. It can be determined that the relationship between the gas escape quantity ($Q$) and filling strip spacing ($x$) and width ($y$) described in Equation (4) could represent the general gas escape quantity of underground mines in northwest China with the overburden composed of loose layer, weakly cemented layer and bedrock layer from top to bottom.

## 4. Conclusions

In this paper, geological and environmental challenges encountered during coal mining and power generation in large coal power bases were considered as a whole, the bearing strength of gray brick after carbonation curing for backfill mining and the stability of the overlying strata after strip backfilling for the geological storage of carbon dioxide were investigated, and the main conclusions were as follows:

(1) After carbonization curing, the strength of gray brick is significantly improved. The uniaxial compressive strengths of the backfill gray brick after 16 days of autoclave curing, carbon dioxide curing and autoclave–carbon dioxide curing are 1.58 MPa, 2.58 MPa, and 9.65 MPa, respectively, reflecting increases of 26%, 98% and 668%, respectively, compared with natural curing. All bricks show X-shape conjugate shear failure. The main hydration

product of the backfill gray brick under carbonization curing conditions is $CaCO_3$, as opposed to $Ca(OH)_2$, which is found under non-carbonization curing conditions. In addition, the internal gaps and pores of the backfill gray brick following carbonization curing are filled by interlaced and needle shaped $CaCO_3$, resulting in higher compactness and integrity.

(2) The stability of overburden is obviously improved by gray brick strip filling. UDEC numerical simulation results show that the maximum surface subsidence reached a maximum of 0.675 m when the filling spacing was 35 m and the width of the filling strip was 11 m, and that it reached a minimum of 0.403 m when the filling spacing was 10 m and the width of the filling strip was 16 m. When the filling spacing was 35 m and the filling strip width was 11 m, the overburden fracture height reached a maximum of 149 m. When the filling spacing was 10 m and the filling strip width was 16 m, the overburden fracture height reached a minimum of 24 m. The surface subsidence, fracture extension and gas tightness of the overlying strata could be improved gradually by increasing the width and reducing the spacing of the brick filling strips.

The results of this paper could provide effective references for green mining and low-carbon utilization of the coal resources in large coal power bases in other countries around the world.

**Author Contributions:** Conceptualization, Z.Z.; methodology, Z.Z.; validation, H.L. and Q.Z.; formal analysis, H.S.; investigation, Z.Z. and H.L.; data curation, H.L. and H.S.; writing—original draft preparation, H.L.; writing—review and editing, Z.Z.; funding acquisition, Z.Z. All authors have read and agreed to the published version of the manuscript.

**Funding:** This research was funded by National Natural Science Foundation of China (51964042), Tianshan Youth Project of Xinjiang Autonomous Region (2019Q063).

**Data Availability Statement:** Not applicable.

**Acknowledgments:** We acknowledge the administrators and technicians in the Zhundong 2# mine for their contributions in the sampling of aeolian sand and fly ash. We appreciate Martha Evonuk, for his generous help in proofreading the article.

**Conflicts of Interest:** The authors declare no conflict of interest.

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
