# Peer review of "Green Mining Takes Place at the Power Plant"

_minerals, doi:10.3390/min12070839_

Round 1

Reviewer 1 Report

Authors have presented one of the modes for green mining and low-carbon utilization of coal resources by studying coal power bases in western China. The basic structure of the paper is scientifically sound, however, the analysis part seems a bit less comprehensive.

It is advisable to add more text to explain the regression part in section 3.3.  Currently, only one parameter is given to explain goodness of fit, which is R2. It is always beneficial to add more parameters like MSE, RMSE etc with explanation to make the idea of fitting more understandable.

It is also advised to add a sub-section to explain the role of sensitive/robust nature of the opted numerical parameters in eqs.(1) and (2). This analysis is necessary to understand the base of the opted equation under specific scenarios, how and why such equation behaves in particular way. 

Table 4 have to be presented in better way, the fracture is a point of interest in the figure, but the way it is presented is hardly comprehensive for readers. it is advisable to add a zoom-in version of cracks. 

Author Response

Thank you for your valuable evaluation for this paper, which is very helpful to improve the quality of this paper. The detailed modifications are as follows:

Point 1: It is advisable to add more text to explain the regression part in section 3.3.  Currently, only one parameter is given to explain goodness of fit, which is R2. It is always beneficial to add more parameters like MSE, RMSE etc with explanation to make the idea of fitting more understandable.

Response 1: First of all, we thank you for your comments, based on your revisions we have added MSE and RMSE parameters on the basis of the original fitting, and explained and analyzed them.

Point 2:It is also advised to add a sub-section to explain the role of sensitive/robust nature of the opted numerical parameters in eqs.(1) and (2). This analysis is necessary to understand the base of the opted equation under specific scenarios, how and why such equation behaves in particular way. 

Response 2: This is a very important comment, and we have added a new paragraph below in eqs.(1) and (2),to analyze the fitted equation, including its characteristics and uses. Meanwhile, we have also analyzed what other engineering geological conditions the parameters in the formula may be subjected to, and intend to continue the research in the future.

Point 3:Table 4 have to be presented in better way, the fracture is a point of interest in the figure, but the way it is presented is hardly comprehensive for readers. it is advisable to add a zoom-in version of cracks. 

Response 3:Thank you for your important comments, and we have optimized the arrangement of the pictures in the article, and made the pictures bigger to ensure that the development of cracks is clearly visible.

Thank you again for your careful review and valuable comments, and best wish to you. We also appreciate Martha Evonuk, PhD, for his generous help in modifying the language of the article.

Reviewer 2 Report

Dear Authors

Just a few comments to be considered.

1.) From my point of view, the title of your contribution might be changed.

1.1.) In the title, the authors note two processes: the extraction and use of low-carbon coal. In this case, the union "and" is used to combine both of these processes into a single chain. But the title is confusing when you read it. It suggests that green mining takes place at the power plant.

1.2.) As I noted in my preface: the work is very relevant. The technology proposed by the authors is applicable not only in the North-West of China. Similar problems exist in all countries. Therefore, regardless of where the experiments were carried out, this technology is applicable in any coal-mining region of the world. From my point of view, the geographical reference in the title underestimates the fundamental nature of the study and reduces the likelihood of citation in the future.

2.) Despite the fact that the presented abstract reflects the essence of the study, from my point of view, it is a little blurry and framed incorrectly. The abstract should clearly indicate the purpose of the study, its importance for society (that is, characterize the problem), indicate the methods and materials of the study. In the abstract presented by the authors, the conclusions are duplicated in detail. Such duplication is not desirable.

2.1) Therefore, it is desirable to avoid narrative text in the annotation.

2.2) Try to use words and phrases: an analysis was made; carried out; studied; developed; proposed; installed and others. It is advisable to start sentences in the abstract with these words and phrases.

2.3 At the end of the abstract, it is necessary to indicate the final result obtained by the authors, for example: A model has been developed that allows ...; Installed a dependency that is....; An effective system (technology) is proposed, and so on.

3.) The manuscript has a rather meager list of references (29 in total), the main volume is made up of Chinese authors. Undoubtedly, in recent times, Chinese scientists have been doing a lot of research and presenting it on display. But the contribution of scientists from other countries is also undeniable. Most of the works are over 5 years old (19) and many are over 10 years old. That creates very weak “citation geography”. The question posed in the present study is very relevant. You have undeservedly overlooked the studies of European, Canadian, Australian, Ukrainian, East European and Russian scientists working in this direction for the last 5 years. I would like to note that research in the field of reducing the ecological impact of mining enterprises, the use of man-made waste for the preparation of backfill mixture and subsequent use in backfill technology, the study of the stress-strain state of the rock mass (the impact of mining on the undermined massif, the aquifer, the preservation of the water-protective layer) was also widely present in the works indexed in Scopus and/or accessible in open-access sources available through MDPI search engine. I wouldn’t dare to suggest you any references as mandatory for your study, I can provide you with some recent contributions devoted to the study of green mining, dynamic manifestations, managing the state of a disturbed array and reducing the risks of these manifestations that I collected for my activities.

Greiving, S.; Gruehn, D.; Reicher, C. The Rhenish Coal-Mining Area—Assessing the Transformational Talents and Challenges of a Region in Fundamental Structural Change. Land 2022, 11, 826. https://doi.org/10.3390/land11060826 (from Germany)

Lim, B.; Alorro, R.D. Technospheric Mining of Mine Wastes: A Review of Applications and Challenges. Sustain. Chem. 2021, 2, 686-706. https://doi.org/10.3390/suschem2040038 (from Australia)

Rybak J., Khayrutdinov M. M., Kuziev D. A., Kongar-Syuryun P. B., & Babyr N. V. Prediction of the geomechanical state of the rock mass when mining salt deposits with stowing. Journal of Mining Institute 2022, 253, 61-70. https://doi.org/10.31897/PMI.2022.2 (from Russia)

4.) This disadvantage is related to the previous one. Due to the fact that the authors did not familiarize themselves with the work of scientists from all over the world,

4.1) they in the introduction (lines 33-49) analyze the problems that arise in the mining and energy sector in China. However, all the described problems are typical for similar enterprises around the world. The lack of an analysis of the typical problems that arise at enterprises around the world, similar to the problems of the Chinese industrial sector, reduces the relevance and fundamental nature of this study. We strongly recommend that the authors include studies by scientists from other countries in the review.

4.2. (lines 50-81) the authors analyze the application of technology with a backfill. As in the case of the analysis of existing problems, it is mainly limited to research, application and implementation only in China. This is not true. The technology with a hardening backfill developed on the basis of man-made waste has been used since the middle of the last century. For the first time, a mixture of classified tailings with cement began to be tested in 1969 at the Mount Isa mine (Australia). This technology has been successfully applied in many mines in Canada and Sweden. In Russia (USSR), tailings were first used to prepare a monolithic backfill in 1969 at the Riddersky mine. In 1970, one chamber was laid at the Gaisky mine (USSR). Currently, man-made wastes are widely used all over the world in the production of backfill composite. Technogenic waste replaces not only the inert filler, but also the binder. Thus, the authors, giving an analysis of the technology used, narrowed down the significance and relevance of this study. Need to be supplemented.

5.) Who made Figures 1 and 2 (?), if by the authors, then it is necessary to indicate (done by the authors), if borrowed, then it is necessary to indicate the source of the borrowing or make an appropriate link to the source. If you have permission to submit a borrowed drawing (without specifying the source), provide this permission to the editorial office.

6.) At the end of the introduction, there are no conclusions on the analysis. This conclusion allows us to characterize the actual question posed, the purpose of the study and the tasks to be solved to achieve this goal. For example: Analyzing the above, it can be noted that ...... is a very topical issue. Therefore, the purpose of this study is ..... and to achieve which it is necessary to solve the following tasks: 1); 2); ..... Such a conclusion allows the reader to understand the vector of the study, and the authors at the end of the study correctly formulate conclusions on it.

7.) I would recommend avoiding group links [13-16]. From my point of view, [11-12] or [17, 21] can be acceptable, up to three are acceptable, everything more needs to be deciphered. Each work you cite is unique and the studies you cite deserve more proper and careful review to demonstrate (and prove) its importance to current research. It is necessary to demonstrate in detail the essence of each study and their need for your work.

8.) Need to expand the methods section

8.1. there is no information on the method of preparation of the samples, namely: how the mixing took place (order, what was poured after), the equipment on which the mixing took place and the speed of mixing. In addition, there are components in the composition, the content of which is unequal, in connection with this I have a question: how was their uniform distribution achieved in the entire mass of the composite

8.2. there is no information under what conditions they were stored (temperature, humidity, pressure in the autoclave, etc.). To what extent these storage conditions correspond to the curing conditions in natural conditions. As I understood from the study, all the test samples were stored for different times and the strength gain was considered in dynamics from time to time: 4-8-12-16 days. Why exactly such a curing time is taken for research. There is no link to the methodology. Why the standard method was excluded: 7-28-60-90. How is your method better than the standard one?

8.3. There is no reference to the method of preparation and testing or similar works, where there is a description of this method, taking into account international experience.

8.4. the authors did not indicate whether the strength characteristics are sufficient for further use and under what conditions. It is not indicated on what equipment the study of physical and mechanical properties was carried out. You must specify the loading speed and time.

8.5. How was convergence achieved?

8.6. the methods should provide a description of the microanalysis methods used, including a description of the tools used and the working conditions. What electron microscope was used? What equipment for x-ray phase analysis was used. What was the pressure in the chamber of the device at the time of the study. The necessary information for disclosure can go on and on. It is necessary to be able to repeat your research by other scientists.

9.) What software for numerical simulation was used?

10.) In the conclusion (1), the authors give well-known facts, these stages of destruction go through all materials obtained by curing mixtures based on binders.

11.) The conclusions are not specific and very vague. This is due to remark (6), the authors did not formulate goals and objectives. In this regard, from my point of view, they incorrectly interpreted the results of their research and their significance.

Summary. From my point of view, the authors have carried out a large research work presented in this manuscript. The study touches on a fairly relevant and interesting topic. From my point of view, again, the manuscript is worthy of publication in the open press with the correction of the shortcomings indicated in my review.

Dear Authors

I appreciate your work. I marked major revision just because of the number of my comments.

Best regards

Author Response

Response to Reviewer 2 Comments

Thank you for your valuable evaluation for this paper, which is very helpful to improve the quality of this paper. The detailed modifications are as follows:

Point 1: From my point of view, the title of your contribution might be changed.

1.1.) In the title, the authors note two processes: the extraction and use of low-carbon coal. In this case, the union "and" is used to combine both of these processes into a single chain. But the title is confusing when you read it. It suggests that green mining takes place at the power plant.

1.2.) As I noted in my preface: the work is very relevant. The technology proposed by the authors is applicable not only in the North-West of China. Similar problems exist in all countries. Therefore, regardless of where the experiments were carried out, this technology is applicable in any coal-mining region of the world. From my point of view, the geographical reference in the title underestimates the fundamental nature of the study and reduces the likelihood of citation in the future.

Response 1:First of all, we thank you for your review and the comments, based on your revisions we have changed the paper title to Green mining takes place at the power plant.

Point 2: Despite the fact that the presented abstract reflects the essence of the study, from my point of view, it is a little blurry and framed incorrectly. The abstract should clearly indicate the purpose of the study, its importance for society (that is, characterize the problem), indicate the methods and materials of the study. In the abstract presented by the authors, the conclusions are duplicated in detail. Such duplication is not desirable.

2.1) Therefore, it is desirable to avoid narrative text in the annotation.

2.2) Try to use words and phrases: an analysis was made; carried out; studied; developed; proposed; installed and others. It is advisable to start sentences in the abstract with these words and phrases.

2.3) At the end of the abstract, it is necessary to indicate the final result obtained by the authors, for example: A model has been developed that allows ...; Installed a dependency that is....; An effective system (technology) is proposed, and so on.

Response 2:First of all, thank you for your review and the comments given, which is an important suggestion. In the process of writing the summary, we may indeed have some problems. Based on your revisions we have rewrote the abstract, as shown below:

Abstract: The number of large coal power plants, characterized by pithead plants, is increasing rapidly in major coal mining countries around the world. Overburden movement caused by coal mining and greenhouse gas emission caused by coal thermal power generation are intertwined, which has become an important challenge for mine ecological environment protection at present and in the future. In order to provide more options for green mining in large coal power plants, a large coal power base in northwest China was took as the researching background in this paper, and a green mining model by both considering the above two aspects of ecological environment damages was proposed, that is, the carbon dioxide greenhouse gas produced by coal-fired power plants can be geologically trapped in goaf, whose overburden stability is controlled by backfill strips made of solid mine waste. In order to explore the feasibility of this model, the bearing strength of the filled grey brick consist mainly of aeolian sand and fly ash under different curing methods was firstly studied, and founded that the strength of the grey brick after carbonization curing was significantly improved. After that, X-ray diffraction (XRD) and Scanning electron microscopy (SEM) were employed to compare the mineral composition and its spatial morphology in grey brick before and after carbonization, and it is believed that the formation of dense acicular calcium carbonate after carbonization curing was the fundamental reason for the improvement of its bearing strength. Finally, a series of stope numerical models were established with UDEC software to analyze the surface settlement, crack propagation height and air tightness of the overlying strata, respectively, when goaf was supported by the backfilling strips with carbonized grey brick. Research results of this paper showed that the stability of overlying strata in goaf can be effectively controlled through adjusting the curing methods, width and spacing of the filled grey brick, so as to facilitate the following geological sequestration of carbon dioxide greenhouse gas in goaf. Consequently the ecological environment damages caused by coal mining and utilization in large coal power base can be resolved as a whole, and the purpose of green mining can be achieved as desired.

Point 3: The manuscript has a rather meager list of references (29 in total), the main volume is made up of Chinese authors. Undoubtedly, in recent times, Chinese scientists have been doing a lot of research and presenting it on display. But the contribution of scientists from other countries is also undeniable. Most of the works are over 5 years old (19) and many are over 10 years old. That creates very weak “citation geography”. The question posed in the present study is very relevant. You have undeservedly overlooked the studies of European, Canadian, Australian, Ukrainian, East European and Russian scientists working in this direction for the last 5 years. I would like to note that research in the field of reducing the ecological impact of mining enterprises, the use of man-made waste for the preparation of backfill mixture and subsequent use in backfill technology, the study of the stress-strain state of the rock mass (the impact of mining on the undermined massif, the aquifer, the preservation of the water-protective layer) was also widely present in the works indexed in Scopus and/or accessible in open-access sources available through MDPI search engine. I wouldn’t dare to suggest you any references as mandatory for your study, I can provide you with some recent contributions devoted to the study of green mining, dynamic manifestations, managing the state of a disturbed array and reducing the risks of these manifestations that I collected for my activities.

Greiving, S.; Gruehn, D.; Reicher, C. The Rhenish Coal-Mining Area—Assessing the Transformational Talents and Challenges of a Region in Fundamental Structural Change. Land 2022, 11, 826. https://doi.org/10.3390/land11060826 (from Germany)

Lim, B.; Alorro, R.D. Technospheric Mining of Mine Wastes: A Review of Applications and Challenges. Sustain. Chem. 2021, 2, 686-706. https://doi.org/10.3390/suschem2040038 (from Australia)

Rybak J., Khayrutdinov M. M., Kuziev D. A., Kongar-Syuryun P. B., & Babyr N. V. Prediction of the geomechanical state of the rock mass when mining salt deposits with stowing. Journal of Mining Institute 2022, 253, 61-70. https://doi.org/10.31897/PMI.2022.2 (from Russia)

Response 3: Thank you for your important comments, based on your revisions we have increased the number of references (39 in total). And we have adjusted the frame of the references in this paper. Now 80% of the references cited from the last five years (32 in total) and about 50% of the references cited from other countries except China.

Point 4:This disadvantage is related to the previous one. Due to the fact that the authors did not familiarize themselves with the work of scientists from all over the world,

4.1) they in the introduction (lines 33-49) analyze the problems that arise in the mining and energy sector in China. However, all the described problems are typical for similar enterprises around the world. The lack of an analysis of the typical problems that arise at enterprises around the world, similar to the problems of the Chinese industrial sector, reduces the relevance and fundamental nature of this study. We strongly recommend that the authors include studies by scientists from other countries in the review.

4.2. (lines 50-81) the authors analyze the application of technology with a backfill. As in the case of the analysis of existing problems, it is mainly limited to research, application and implementation only in China. This is not true. The technology with a hardening backfill developed on the basis of man-made waste has been used since the middle of the last century. For the first time, a mixture of classified tailings with cement began to be tested in 1969 at the Mount Isa mine (Australia). This technology has been successfully applied in many mines in Canada and Sweden. In Russia (USSR), tailings were first used to prepare a monolithic backfill in 1969 at the Riddersky mine. In 1970, one chamber was laid at the Gaisky mine (USSR). Currently, man-made wastes are widely used all over the world in the production of backfill composite. Technogenic waste replaces not only the inert filler, but also the binder. Thus, the authors, giving an analysis of the technology used, narrowed down the significance and relevance of this study. Need to be supplemented.

Response 4: First of all, we thank you for your comments, based on your revisions we have added studies by scientists from other countries to analyze the typical problems that arise at enterprises around the world. And we have supplemented the studies on the  technology with a hardening backfill developed on the basis of man-made waste. The modified section is shown below:

For several decades coal has significantly contributed to the global energy needs, accounting for 25% of the global energy production in 2000, 30% in 2010, and 27% in 2020 [1]. However, in the process of coal mining, it often brings about serious overburden movement, which causes the loss of groundwater resources and surface collapse, changes the soil structure and damages the ecological environment seriously [2-3]. For example, in China, the area of land destroyed by mining has reached 2 million hm2 [4], in which the area of settlement land has reached 1 million hm2 [5]. These mining impacts pose significant environmental, socio-economic and mining layout challenges. For this reason, these adverse impacts and mitigation measures have been extensively studied in several countries, including Russia [6], Australia [7-8], the United Kingdom [9], South Africa [10-11], India [12-13] and Germany [14]. Meanwhile, The number of large coal power plants, characterized by pithead plants, is increasing rapidly in major coal mining countries around the world. Overburden movement caused by coal mining and greenhouse gas emission caused by coal thermal power generation are intertwined, which has become an important challenge for mine ecological environment protection at present and in the future, as shown in figure 1.

Figure 1. Ecological and environmental effects of the traditional coal exploitation and utilization mode in large coal power bases

To resolve the problems of the environmental damage caused by the mining and utilization of coal as a fossil fuel in large coal power bases, an innovative mode of green mining and low-carbon utilization of the coal resources is proposed, as shown in Figure 2. Firstly, the aeolian sand abundant on the surface of mining area and fly ash produced by thermal power plant are used as the main raw materials to make the embryo body of filling grey brick. Then, the grey brick was curved with carbon dioxide gas from thermal power plant, and backfilled in underground gob to support overlying strata. Finally, the carbon dioxide gas can be injected and stored in gob.

Figure 2. Innovative mode of green mining and low-carbon utilization for coal resources in large coal power bases

In the past decades, backfilling has been extensively applied in underground coal mining to resolve safety and environmental problems. The backfill bodies were employed to reduce ground subsidence [15-16], underground water loss [17-18], and so forth. Meanwhile, backfill mining employs coal gangue (CG), fly ash (FA), and tailings as primary materials to provides a feasible approach for treatment and utilization of solid wastes from mining industries [19-20]. On this basis, many valuable studies have been carried out recently. For the first time, a mixture of classified tailings with cement began to be tested in 1969 at the Mount Isa mine in Australia. This technology has been successfully applied in many mines in Canada and Sweden. In Russia (USSR), tailings were first used to prepare a monolithic backfill in 1969 at the Riddersky mine. In 1970, one chamber was laid at the Gaisky mine (USSR). Currently, man-made wastes are widely used in the production of backfill composite around the word. Technogenic waste replaces not only the inert filler, but also the binder. Ercikdi et al used waste glass and silica fumes as artificial pozzolanas to prepare tailings cementing filling materials and investigated the effect of the type of waste materials on the early strength and later strength of the filled object [21-22].Feng et al studied the mechanical properties of cemented filling materials in which gangue was partially replaced by waste concrete[23,28]. Peyronnard and Benzaazoua studied the effect of CAlSiFrit and deinking sludge fly ash as partial binder replacement in cemented paste backfill (CPB)[24]. Cihangir et al used alkali-activated neutral and acidic blast furnace slags (AASs) with aqueous sodium silicate (LSS), and sodium hydroxide (SH) were tested as alternative binders to OPC for CPB of high-sulphide mill tailings[25] .Deng et al developed a new type of cemented filling material, using waste rock as a coarse aggregate, FA as a fine powder, slag as an activator, and ordinary Portland cement as a binder[26,28]. Xu et al studied the strength development and microstructure evolution of cemented tailings backfill containing different binder types and contents[27]. Zhou et al. explored the feasibility of replacing cementitious filling materials with air-accumulated sand as aggregate, and investigated the effects of fly ash (FA) content, cement content, lime slag (LS) content and concentration on the mechanical properties of air-accumulated sand-based cementitious filling materials [28]. Wang et al. used hydrogen peroxide (H2O2) as a chemical blowing agent to improve the foaming performance of the cemented foam and the reinforcing effect of the foam based on the cemented foam with gangue and fly ash as the main raw materials [29]. Ermolovich et al studied the possibility of creating and using nanomodified backfill material based on the waste from enrichment of water-soluble ores [30].

Point 5:Who made Figures 1 and 2 (?), if by the authors, then it is necessary to indicate (done by the authors), if borrowed, then it is necessary to indicate the source of the borrowing or make an appropriate link to the source. If you have permission to submit a borrowed drawing (without specifying the source), provide this permission to the editorial office.

Response 5: The ideas for making Figure 1 and 2 were entirely created by our research team and made by the author of this paper

Point 6:At the end of the introduction, there are no conclusions on the analysis. This conclusion allows us to characterize the actual question posed, the purpose of the study and the tasks to be solved to achieve this goal. For example: Analyzing the above, it can be noted that ...... is a very topical issue. Therefore, the purpose of this study is ..... and to achieve which it is necessary to solve the following tasks: 1); 2); ..... Such a conclusion allows the reader to understand the vector of the study, and the authors at the end of the study correctly formulate conclusions on it.

Response 6: This is a very important comment, and we have modified the last part of the introduction ,as shown below:

Analyzing the above, it can be noted that reducing or ablating the damage of ecological environment caused by mining activities by green mining is a very topical issue. With the centralized mining and utilization of coal resources, related environmental problems will appear at the same time, which provides a possibility for the comprehensive solution of various problems. Therefore, the purpose of this study is exploring a green mining model that can simultaneously control the strata stability over goaf and geologic sequestration of carbon dioxide greenhouse gas in goaf, and to achieve which it is necessary to solve the following tasks: 1) How to prepare the backfilling material in goaf with certain mine solid waste, and make it have the bearing strength to meet the requirements through appropriate curing method; 2) How to evaluate the stability of the overlying strata after goaf is backfilled by the above designed materials.

Point 7:I would recommend avoiding group links [13-16]. From my point of view, [11-12] or [17, 21] can be acceptable, up to three are acceptable, everything more needs to be deciphered. Each work you cite is unique and the studies you cite deserve more proper and careful review to demonstrate (and prove) its importance to current research. It is necessary to demonstrate in detail the essence of each study and their need for your work.

Response 7: Thank you very much. You are very careful. I have corrected the problem and there are no more consecutive quotes of three or more papers in this paper.

Point 8:Need to expand the methods section

8.1. there is no information on the method of preparation of the samples, namely: how the mixing took place (order, what was poured after), the equipment on which the mixing took place and the speed of mixing. In addition, there are components in the composition, the content of which is unequal, in connection with this I have a question: how was their uniform distribution achieved in the entire mass of the composite

Response 8.1:First of all, thank you for your review and the comments given, which is an important issue. In the research methods section, we have supplemented and analyzed the chemical composition of each raw material in the experiments, and the results are shown in the figure below. The main composition of quicklime is CaO, but also contains a small amount of SiO2 and MgO, its CaO content is up to 89%. The main chemical composition of gypsum are CaO and SO3, and also contains a small amount of SiO2, Al2O3 and Fe2O3.The fly ash was primarily composed of mullite and quartz, and the SiO2 content is 55%, Al2O3 content is 25%.

(a) Chemical element content of Quicklime   (b) Chemical element content of Gypsum

(c)Chemical element content of Fly ash

Based on your revisions we have supplemented the information on how the mixing took place (order, what was poured after), the equipment on which the mixing took place and the speed of mixing in this paper, as shown below.

In this study, the ratio of water: cement was 2:3, and the ratio of gypsum: fly ash: quicklime: anthracite sand was 1:2:3:12.The dry Ingredients were mixed and thoroughly homogenized using a blender (HJW-60 series) for 3 min before water was added into the mixture and then stirred for another 2 min. After mixing, the mixed slurry was cast into an iron mold with dimensions of 5 cm × 5 cm × 5 cm and three specimens were made for each curing age using different curing methods to reduce the test error. The specimens were then subjected to natural, autoclave, carbon dioxide and autoclave-carbon dioxide curing for 4 days, 8 days,12 days and 16 days[31],as shown in Table 1. Finally, uniaxial compressive strength(UCS) tests, carbonization tests, X-ray diffraction analysis (XRD) tests and scanning electron microscope (SEM) tests were carried out on the specimens.

8.2. there is no information under what conditions they were stored (temperature, humidity, pressure in the autoclave, etc.). To what extent these storage conditions correspond to the curing conditions in natural conditions. As I understood from the study, all the test samples were stored for different times and the strength gain was considered in dynamics from time to time: 4-8-12-16 days. Why exactly such a curing time is taken for research. There is no link to the methodology. Why the standard method was excluded: 7-28-60-90. How is your method better than the standard one?

Response 8.2:The information under what conditions they were stored (temperature, humidity, pressure in the autoclave, etc.) is explained in the curing methods on the right side of Table 1 in this paper. The temperature and pressure of autoclaved curing were 130°C and 0.165MPa, respectively. And we set the curing time as 4 days,8 days,12days and 16 days according to this reference.(Bai,X.C.;Duan,Z.H. Development of the Light Quality and High Strength Fly Ash Brick. Journal of Henan University(Natural Science) 2009,39,5.)

8.3. There is no reference to the method of preparation and testing or similar works, where there is a description of this method, taking into account international experience.

Response 8.3: Based on your revisions we have included relevant references in this section.

8.4. the authors did not indicate whether the strength characteristics are sufficient for further use and under what conditions. It is not indicated on what equipment the study of physical and mechanical properties was carried out. You must specify the loading speed and time.

Response 8.4: Based on your revisions we have supplemented this section, as shown below.

The UCS tests of grey brick were carried out using the SANS brand hydraulic single shaft compressor in the Mechanics Laboratory, School of Mechanical Engineering, Xinjiang University , China as per the Chinese standard (JGJ/T 70-2009) The maximum test force of uniaxial compressor is 300KN, and the accuracy of force value is below ±0.3%. According to the standard, a displacement loading model was used to avoid specimens rapidly breaking[32]. In this way, the whole stress-strain curve was obtained. In this study, the pre-peak loading speed was 0.1 mm/s and the loading speed after peak was 0.2 mm/s.

8.5. How was convergence achieved?

Response 8.5:We used the Origin software to analyze and fit the data.

8.6. the methods should provide a description of the microanalysis methods used, including a description of the tools used and the working conditions. What electron microscope was used? What equipment for x-ray phase analysis was used. What was the pressure in the chamber of the device at the time of the study. The necessary information for disclosure can go on and on. It is necessary to be able to repeat your research by other scientists.

Response 8.6:Thank you for your important comments, and we have supplemented this part in this paper, as shown below.

The XRD tests were carried out with D8 Advance X-ray powder diffractometer (Bruker AXS GmbH). The radiation source was Cu target, the scanning range 2θ was 5°~ 80°, the scanning speed was 10°/min, the experimental voltage was 60kV, and the test current was 50mA.The SEM tests were carried out by LEO-1430VP scanning electron microscope (Zeiss, Germany) with the magnification of 50~20000 times. Carbonation depth of grey brick under different curing methods was detected by phenolphthalein alcohol method (phenolphthalein alcohol solution with mass fraction of 1% as chromogenic agent) [33], and carbonation depth was measured by digital carbonation depth scale (China Zhuolin Science and Technology), with measuring accuracy of 0.01mm and measuring range of 0-25mm.

Point 9:What software for numerical simulation was used?

Response 9:UDEC was used for numerical simulation in this paper and it was briefly introduced in this paper. UDEC is a two-dimensional numerical calculation program based on continuum simulation discrete element, which mainly simulates the mechanical behavior of discontinuous medium (such as joint block) under static or dynamic load conditions. UDEC discontinuous media is reflected by the combination of separated blocks, and the joints are treated as boundary conditions between blocks, allowing blocks to move and turn along the joint surface[34]. UDEC can clearly simulate the development of cracks in overlying strata during grey brick backfill mining, so as to achieve the desired simulation effect.

Point 10:In the conclusion (1), the authors give well-known facts, these stages of destruction go through all materials obtained by curing mixtures based on binders.

Response 10:Based on your revisions we have deleted the analysis of deformation and failure stage of grey brick in the final conclusion.

Point 11: The conclusions are not specific and very vague. This is due to remark (6), the authors did not formulate goals and objectives. In this regard, from my point of view, they incorrectly interpreted the results of their research and their significance.

Response 11:At the end of the paper, the conclusions are summarized again, and the new conclusions are shown as follows.

In this paper, geological and environmental challenges encountered during coal mining and power generation in large coal power bases were considered as a whole, and the bearing strength of grey brick after carbonation curing for backfill mining and the stability of the overlying strata after strip backfilling for the geological storage of carbon dioxide are investigated, and the main conclusions were drawn as follows:

(1) After carbonization curing, the strength of grey brick is significantly improved. The uniaxial compressive strengths of the backfill grey brick after 16 days of autoclave curing, carbon dioxide curing and autoclave-carbon dioxide curing are 1.58 MPa, 2.58 MPa, and 9.65 MPa, respectively, reflecting increases of 26%, 98%, and 668%, respectively, compared with natural curing. All brick show X-shape conjugate shear failure. The main hydration product of the backfill grey brick under carbonization curing conditions is CaCO3, as opposed to Ca(OH)2, which is found under non-carbonization curing conditions. In addition, the internal gaps and pores of the backfill grey brick following carbonization curing are filled by interlaced and needle-shaped CaCO3, resulting in higher compactness and integrity.

(2) The stability of overburden is improved obviously with grey brick strip filling. UDEC numerical simulation results show that, the maximum surface subsidence reached a maximum of 0.675 m when the filling spacing was 35 m and the width of the filling strip was 11 m and that it reached a minimum of 0.403 m when the filling spacing was 10 m and the width of the filling strip was 16 m. when the filling spacing is 35 m and the filling strip width is 11 m, the overburden fracture height reaches a maximum of 149 m and that, when the filling spacing is 10 m and the filling strip width is 16 m, the overburden fracture height reaches a minimum of 24 m. The surface subsidence, fracture extension and gas tightness of the overlying strata can be improved gradually by increasing the width and reducing the spacing of the brick filling strips. The results of this paper could provide effective references for the green mining and low-carbon utilization of the coal resources in large coal power bases in other countries around the world.

Thank you again for your careful review and valuable comments, and best wish to you.

Round 2

Reviewer 1 Report

Most of the minor comments are addressed by the reviewers. However, the only major concern I pointed out still lacking, i.e. explaining the role of the sensitive/robust nature of the opted numerical parameters in eqs.(1) and (2). This analysis is necessary to understand the base of the opted equation under specific scenarios, and how and why such equation behaves in a particular way. In my honest view, the sensitive/robust analysis is a part of regression analysis and must be reported within the article. 

Author Response

Response to Reviewer 1 Comments:

Dear reviewer, thank you for your valuable evaluation for this paper, which is very helpful to improve the quality of this paper. The detailed modifications are as follows:

Point 1: It is also advised to add a sub-section to explain the role of sensitive/robust nature of the opted numerical parameters in eqs.(1) and (2). This analysis is necessary to understand the base of the opted equation under specific scenarios, how and why such equation behaves in particular way.

Response 2: Dear reviewer, a sub-section to explain the role of sensitive nature of the opted Equation (1) and (2) are added in manuscript, and the relative content is modified further to illustrate the base and behaving characteristics of the Equations more clearly. The detailed modifications are as follows:

The simulation results showed that the maximum surface subsidence always occurred directly above the gob, and this subsidence under mining with different filling spacing and filling strip widths are shown in Figure 11. It can be seen that the maximum surface subsidence reached a maximum of 0.675 m when the filling spacing was 35 m and the width of the filling strip was 11 m and that it reached a minimum of 0.403 m when the filling spacing was 10 m and the width of the filling strip was 16 m. The maximum surface subsidence could be reduced obviously when the filling spacing decreased from 35 m to 20 m and the filling strip width increased from 11 m to 13. In order to describe the quantitative relationship between the maximum surface subsidence (S) and the filling strip spacing (x) and the filling strip width (y) more accurately, Gauss2D function of the Origin software was used to obtain the fitting relationship between above parameters, as shown in Equation (1). According to the results of fitting calculation, the fitting correlation coefficient (R2) was 0.9926, the mean square error (MSE) was 3.86×10-5, and the root mean square error (RMSE) was 0.0062, and the goodness of this fit was acceptable. According to the Equation (1), the maximum surface subsidence (S) has an exponential relationship with filling strip spacing (x) and filling strip width (y). Furtherly, the maximum surface subsidence (S) decreases continuously with decreasing filling spacing (x) and increasing filling strip width (y), which provides valuable reference for controlling the maximum surface subsidence (S) through adjusting the backfill settings. The primary consideration for the filling strips spacing (x) is to avoid the caving of the immediate roof strata over gob due to fact that the gob must be kept integrated for the purpose of following storage of carbon dioxide gas. The filling strip width (y) is limited commonly by the construction speed of filling strips because the backfilling speed and mining speed need to be consistent to ensure the continuity of backfill mining operations. In practice, the filling strips spacing (x) can be adjusted in a larger range compared with the filling strip width (y) because the fact that the time for backfilling in a backfill mining cycle is always limited. Consequently, it is more feasible to control the maximum surface subsidence (S) by adjusting the filling strip spacing (x). It also should be noted that the parameters except for filling strip spacing (x) and filling strip width (y) in the Equation (1) are only applicable to the engineering geological conditions in this study, and should be adjusted according to the mining depth, rock strata mechanics and backfilling materials’ strength in different cases. It can be determined that the relationship between the maximum surface subsidence (S) and filling strip spacing (x) and width (y) described in the Equation (1) could represent the general surface subsidence of underground mines in northwest China with the overburden composed of loose layer, weakly cemented layer and bedrock layer from top to bottom.

Similar modifications are made for Equation (2) and (4).

Thank you again for your careful review and valuable comments, and best wish to you.

Reviewer 2 Report

Dear Authors

I appreciate the amount of work that you devoted to improve your study.

It was a pleasure to find cautious responses to all of my comments and concerns.

Best regards

Author Response

Thank you again for your careful review and valuable comments, and best wish to you.